# Mortality outcomes with hydroxychloroquine and chloroquine in COVID-19 from an international collaborative meta-analysis of randomized trials

Substantial COVID-19 research investment has been allocated to randomized clinical trials (RCTs) on hydroxychloroquine/chloroquine, which currently face recruitment challenges or early discontinuation. We aim to estimate the effects of hydroxychloroquine and chloroquine on survival in COVID-19 from all currently available RCT evidence, published and unpublished. We present a rapid meta-analysis of ongoing, completed, or discontinued RCTs on hydroxychloroquine or chloroquine treatment for any COVID-19 patients (protocol: https://osf.io/QESV4/). We systematically identified unpublished RCTs (ClinicalTrials.gov, WHO International Clinical Trials Registry Platform, Cochrane COVID-registry up to June 11, 2020), and published RCTs (PubMed, medRxiv and bioRxiv up to October 16, 2020). All-cause mortality has been extracted (publications/preprints) or requested from investigators and combined in random-effects meta-analyses, calculating odds ratios (ORs) with 95% confidence intervals (CIs), separately for hydroxychloroquine and chloroquine. Prespecified subgroup analyses include patient setting, diagnostic confirmation, control type, and publication status. Sixty-three trials were potentially eligible. We included 14 unpublished trials (1308 patients) and 14 publications/preprints (9011 patients). Results for hydroxychloroquine are dominated by RECOVERY and WHO SOLIDARITY, two highly pragmatic trials, which employed relatively high doses and included 4716 and 1853 patients, respectively (67% of the total sample size). The combined OR on all-cause mortality for hydroxychloroquine is 1.11 (95% CI: 1.02, 1.20; $I^2 = 0\%$; 26 trials; 10,012 patients) and for chloroquine 1.77 (95%CI: 0.15, 21.13, $I^2 = 0\%$; 4 trials; 307 patients). We identified no subgroup effects. We found that treatment with hydroxychloroquine is associated with increased mortality in COVID-19 patients, and there is no benefit of chloroquine. Findings have unclear generalizability to outpatients, children, pregnant women, and people with comorbidities.

Coronavirus disease 2019 (COVID-19) caused by severe acute respiratory syndrome coronavirus 2 (SARS-CoV-2) has the potential of progression into respiratory failure and death[1]. More than 1,500,000 persons with COVID-19 globally have died by December 2020[2], and treatment options are limited[3]. The COVID-19 pandemic has caused a hitherto unprecedented search for possible therapies, with almost 700 clinical trials initiated in the first quarter of 2020—and one in five of these trials target hydroxychloroquine (HCQ) or chloroquine (CQ)[4]. This remarkable attention was primarily due to in vitro data[5], immunomodulatory capacities[6], and the oral formulation and well-documented safety profiles.

In March 2020, the US Food and Drug Administration (FDA) issued an Emergency Use Authorization of HCQ[7] and the number of prescriptions and usage outside clinical studies skyrocketed[8]. In many countries, HCQ or CQ were listed in treatment guidelines for COVID-19 (including, e.g., China, Ireland, and the United States)[9]. In a New York City cohort of 1376 COVID-19 inpatients during March–April 2020, 59% received HCQ[10]. However, the FDA revoked the Emergency Use Authorization on June 15, 2020[11]. At that point, two large randomized clinical trials (RCTs), RECOVERY and the WHO Solidarity trial, had stopped enrollment to their HCQ treatment arms[12,13]. An interim analysis of the RECOVERY trial showed no mortality benefit of HCQ[13]. Established as treatments of malaria and rheumatic disorders, HCQ and CQ may carry potentially severe adverse effects, especially related to cardiac arrhythmia[6]. Public uncertainty still remains, as illustrated by recent reports of planned use in pandemic epicenters in Central and South America[14].

While many trials are ongoing, additional published evidence of potential benefits or harms may be several months away, if they even reach completion. Given the lack of favorable results in the large RECOVERY trial and the revoked Emergency Use Authorization, recruitment into HCQ and CQ trials has become increasingly difficult and many trials may run the risk of ending in futility. A rapid examination of data on all-cause mortality from as many trials as possible may offer the best evidence on potential survival benefits and to ensure that patients are not exposed to unnecessary risks if benefit is lacking. We used the infrastructure established with COVID-evidence[15], a comprehensive database of COVID-19 trials funded by the Swiss National Science Foundation, to invite all investigators of HCQ or CQ trials to participate in an international collaborative meta-analysis. We aimed to identify and combine all RCTs investigating the effects of HCQ or CQ on all-cause mortality in patients with COVID-19 compared to any control arm similar to the experimental arm in all aspects except the administration of HCQ or CQ.

In this work, we find that treatment with HCQ is associated with increased mortality in COVID-19 patients, and there is no benefit of CQ. Findings have unclear generalizability to outpatients, children, pregnant women, and people with comorbidities.

## Results

Our search identified 146 randomized trials investigating HCQ or CQ as treatment for COVID-19, of which 83 were deemed potentially eligible after scrutinizing the randomized comparisons. The investigators of these 83 trials were contacted and 57% (47 of 83) responded (Fig. 1). Of the responders, 19 trials were eligible and available (14 unpublished, one preprint, and four publications); 21 trials were ineligible according to information provided; five responding investigator teams were not ready to share their results yet; and two declined participation. For the 36 trials without response, six were confirmed eligible and available

(four publications and two preprints); two were confirmed ineligible; and for the remaining 28, results were not available, nor could they be confirmed eligible.

We included 28 trials (14 unpublished trials, nine publications, and five preprints; of these, one publication and two preprints were identified for the first time in our search update)[13,16–28]. Individual trial characteristics are presented in Table 1 (28 included trials) and Supplement Table S1 (34 potentially eligible but unavailable). Overall, trial characteristics were not different between included and unavailable trials (Table 2).

HCQ was evaluated in 26 trials (10,012 patients) and CQ was evaluated in four trials (307 patients). Two trials investigated both HCQ versus control and CQ versus control (63 patients). The median sample size was 95 (interquartile range (IQR) 28–282) for HCQ trials and 42 (IQR 29–95) for CQ trials. The two largest trials (RECOVERY and WHO SOLIDARITY) included 47% and 19% of all patients in the HCQ trials, respectively. Most trials investigated HCQ or CQ in hospitalized patients (22 trials; 79%), and only five trials (18%) had an outpatient setting. The average mortality was 10.3% (standard deviation 13.5%) in inpatient trials and 0.08% (standard deviation 0.18%) in outpatient trials. The comparator was in 11 trials placebo (39%) and in 17 (61%) no other treatment than standard of care. In most trials, patients and clinicians were aware of the treatment (15 trials; 54%), while in one trial (4%) the patients were blinded and in 11 trials (39%) patients and clinicians were blinded (Table 2). We identified no relevant risk of bias across all trials, with only one trial including seven patients having an overall high risk of bias (Table S2). We found no evidence of small-study effects (Figure S1).

Regarding HCQ, in the 26 included trials, 606 of 4316 (14.0%) patients treated with HCQ died and 960 of 5696 patients (16.9%) in the control groups died (within 19 trials with a 1:1 randomization ratio, 7.7% of patients in the HCQ arm died [181 of 2346] and 7.1% of patients in the control arm died [168 of 2352]). In the meta-analysis, the combined odds ratio (OR) was 1.11 (95% confidence interval (CI), 1.02–1.20, $p = 0.02$), with low heterogeneity ($I^2 = 0\%$) (Fig. 2A). In 12 trials including a total of 1282 patients (representing 12.8% of the total sample size for HCQ), there were zero deaths in both arms.

Regarding CQ, in the four included trials, 18 of 160 (11%) patients treated with CQ died and 12 of 147 patients (8%) in the control groups died. The combined OR was 1.77 (95% CI 0.15–21.13, $p = 0.21$), with low heterogeneity ($I^2 = 0\%$) (Fig. 2B). In two of four trials including a total of 217 patients, there were zero deaths in both arms.

The available evidence in this study is the result of publications, preprints, or unpublished trial results accrued from April 10, 2020 to October 16, 2020 as shown in the cumulative meta-analyses (Fig. 3A–C).

For HCQ, none of the exploratory subgroup analyses showed an effect modification (Supplement Table S3 and Figure S1A, B). When only including published information (publications and preprints, excluding unpublished trials), there was an increase in mortality among patients treated with HCQ (OR 1.12, 95% CI 1.08–1.16), while among the unpublished trials there was no such sign of increased mortality (OR 0.92, 95% CI 0.63–1.34, $p$ for interaction = 0.23). We conducted no subgroup analyses for CQ, as there were only two trials with events. In the sensitivity analyses employing different meta-analytical approaches (Supplement Table S4 and Figures S2A–C), results were compatible with the main analysis.

## Discussion

This collaborative meta-analysis of 28 published or unpublished RCTs, including 10,319 patients, shows that treatment with HCQ

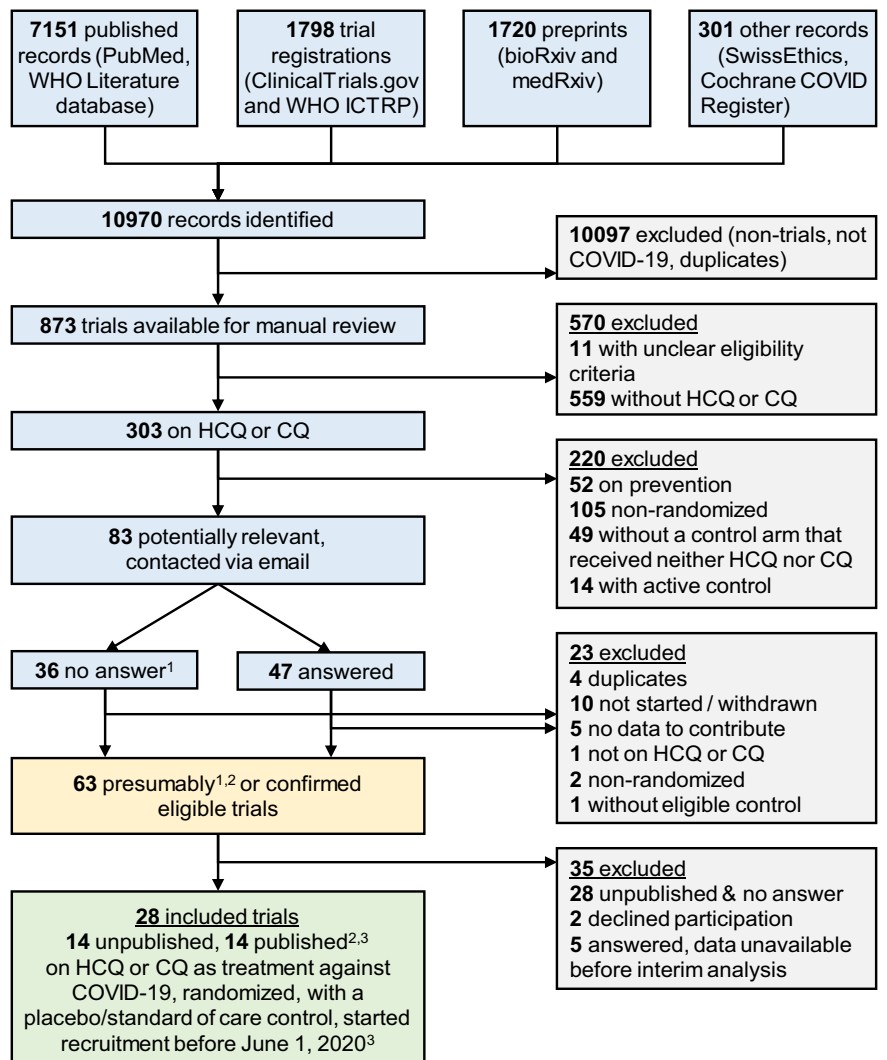

**Fig. 1 Flowchart of included randomized clinical trials.** Sources searched up to June 11, 2020 (PubMed, ClinicalTrials.gov, WHO ICTRP, Cochrane COVID Register) or April 9 (WHO Literature database, bioRxiv, medRxiv, SwissEthics). [1]Trials for which we received no answer were presumed to be eligible unless withdrawn. [2]One publication and two preprints were identified in a later search update. [3]Published peer-reviewed articles or posted preprints. CQ chloroquine, HCQ hydroxychloroquine, ICTRP International Clinical Trials Registry Platform, WHO World Health Organization.

was associated with increased mortality in COVID-19 patients, and there was no benefit from treatment with CQ. No differences were seen across subgroup analyses on patient setting, diagnosis confirmation, control type, publication status, or dose and the between-study heterogeneity was low. For CQ, the number of studies was too small to draw clear conclusions.

This meta-analysis offers useful insights for a challenging health situation. Hundreds of thousands of patients have received HCQ and CQ outside of clinical trials without evidence of their beneficial effects. Public interest is unprecedented, with weak early evidence supporting HCQ's merits being widely discussed in some media and social networks—despite the unfavorable results by a very large RCT. Numerous clinical studies have been investigating HCQ and CQ almost simultaneously. Although several systematic reviews and meta-analyses are already available, they only consider the small handful of RCTs being already published (which were all included here)[29–33]. While data sharing has been rather limited to date in biomedical research, such openness can be transformative in generating knowledge. This pandemic has brought together a collaboration of clinical trialists agreeing to share their data, which allows this study to not only

summarize the existing evidence but also illustrate the accumulation of evidence that would otherwise not be available.

For HCQ, evidence is dominated by the RECOVERY trial[13], which indicated no mortality benefit for treated COVID-19 patients, together with longer hospitalization and higher risk of progression to invasive mechanical ventilation and/or death. Similarly, the WHO SOLIDARITY trial indicated no mortality benefit[26]. The RECOVERY and WHO SOLIDARITY trials used HCQ in comparatively higher doses than all other trials except REMAP-CAP. There was no evidence for an effect modification by dose (p for interaction = 0.29), and the combined effect of all the trials with lower dose did not indicate a benefit of HCQ but tended to a null effect (OR 0.97; 0.73–1.30) with wide CIs, compatible with the main effect estimate.

This meta-analysis does not address prophylactic use nor other outcomes besides mortality. All but three trials excluded children and the majority excluded pregnant or breastfeeding women; generalizability remains unclear for those populations. Among the five studies on outpatients, there were three deaths, two occurring in the one trial of 491 relatively young patients with few comorbidities[17] and one occurring in a small trial with 27

**Table 1 Group-level characteristics of randomized clinical trials evaluating hydroxychloroquine or chloroquine as treatment for COVID-19.**

| Acronym | Register ID | Setting | Treatment comparison Experimental arm (n) | Control arm (n) | Treatment schedule | COVID-19 status | Age | Mortality (%) | Location | Blinding | Targeted sample size | Trial and publication status[a] |
|---|---|---|---|---|---|---|---|---|---|---|---|---|
| **Hydroxychloroquine** | | | | | | | | | | | | |
| REMAP-CAP | NCT02735707 | ICU[b] | HCQ (61) | No treatment (81) | 800 mg at 0 and 6 h, then 800 mg/day for up to 6 days | Confirmed and suspected | ≥18 years | 27.46 | International[c] | None | No fixed sample size | Completed[d], not published |
| | ChiCTR2000029559 | Inpatient | HCQ (31) | No treatment (31) | 200 mg twice a day for 5 days | Confirmed | ≥18 years | 0 | China | Participant, caregiver | 300 | Completed and published |
| | ChiCTR2000029868 | Inpatient | HCQ (75) | No treatment (75) | 1200 mg/day for 3 days, then 800 mg/day for 11–18 days | Confirmed | ≥18 years | 0 | China | None | 360 | Completed and published |
| | NCT04261517 | Inpatient | HCQ (15) | No treatment (15) | 400 mg/day for 5 days | Confirmed | ≥18 years | 0 | China | None | 30 | Completed and published |
| RECOVERY | NCT04381936 | Inpatient | HCQ (1561) | No treatment (31,55) | 800 mg at 0 h, then 800 mg after 6 h, then 800 mg/day for up to 9 days | Confirmed and suspected | ≥18 years | 25.67 | Europe | None | 12,000[e] | Completed[d] and published |
| | ChiCTR2000030054 | Inpatient | HCQ (18) | No treatment (12) | 400 mg/day for 10 days | Confirmed | 18–75 years | 0 | China | None | 100 | Completed and published |
| NO COVID-19 | NCT04316377 | Inpatient | HCQ (27) | No treatment (26) | 800 mg/day for 7 days | Confirmed | ≥18 years | 3.77 | Norway | None | 202 | Halted and published |
| | NCT04384380 | Inpatient | HCQ (21) | No treatment (12) | 800 mg/day for 1 day, then 400 mg/day for 6 days | Confirmed | 20–79 years | 0 | Taiwan | None | 45 | Completed and published |
| | NCT04353336 | Inpatient | HCQ (97) | No treatment (97) | 800 mg/day for 1 day, then 400 mg/day for 14 days | Confirmed | All | 5.67 | Egypt | None | 40 | Recruiting and published |
| Coalition I | NCT04322123 | Inpatient | HCQ (221) | No treatment (227) | 800 mg/day for 7 days | Confirmed and suspected | ≥18 years | 2.90 | Americas | None | 630[e] | Completed and published |
| TEACH | NCT04369742 | Inpatient | HCQ (67) | Placebo (61) | 800 mg/day for 1 day, then 400 mg/day for 4 days | Confirmed | ≥12 years | 10.16 | United States | Participant, caregiver | 626 | Terminated and published |
| | NCT04491994 | Inpatient | HCQ (349) | No treatment (151) | 400 mg twice a day for 1 day, then 200 mg twice a day for 4 days | Confirmed | ≥18 years | 0 | Pakistan | None | 540 | Completed and published |
| WHO SOLIDARITY | 2020-001366-11 | Inpatient | HCQ (947) | No treatment (906) | Day 1, 2000 mg (hour 0, 800 mg; hour 6, 800 mg; hour 12, 400 mg), thereafter 800 mg per day for 10 days | Confirmed | ≥18 years | 10.15 | International | None | 10,000[e] | Terminated and published |
| PATCH | NCT04329923 | Inpatient | HCQ (15) | Placebo (15) | 800 mg/day for up to 14 days | Confirmed | ≥40 years | 0 | United States | Participant, caregiver | 400[e] | Recruiting, not published |
| CCAP-1 | NCT04345289 | Inpatient | HCQ (1) | Placebo (1) | 600 mg/day for 7 days | Confirmed | ≥18 years | 0 | Denmark | Participant, caregiver | 1500[e] | Terminated, not published |
| | NCT04335552 | Inpatient | HCQ (4) / HCQ + azithromycin (2) | No treatment (2) / Azithromycin (3) | 800 mg/day for 1 day, then 600 mg/day for 4 days / 800 mg/day for 1 day, then 600 mg/day for 4 days | Confirmed | ≥12 years | 16.67 / 60 | United States | None | 500 | Terminated, not published |
| ARCHAIC | NL8490 | Inpatient | HCQ (4) | No treatment (3) | 800 mg/day for 1 day, then 400 mg/day for 4 days | Confirmed | ≥18 years | 28.57 | Netherlands | None | 950 | Terminated, not published |
| HYDRA | NCT04315896 | Inpatient | HCQ (75) | Placebo (77) | 400 mg/day for 10 days | Confirmed | 18–80 years | 37.50 | Mexico | None | 500 | Recruiting, not published |
| PROTECT | NCT04338698 | Inpatient | HCQ + oseltamivir (64) / HCQ + oseltamivir (62) / HCQ + azithromycin (59) | Azithromycin + oseltamivir (64) / Oseltamivir (63) / Azithromycin (61) | 600 mg/day for 5 days | Confirmed | ≥18 years | 0 / 0.80 / 2.50 | Pakistan | Investigator | 500 | Recruiting, not published |

**Table 1 (continued)**

| Acronym | Register ID | Setting | Treatment comparison | | Treatment schedule | COVID-19 status | Age | Mortality (%) | Location | Blinding | Targeted sample size | Trial and publication status[a] |
|---|---|---|---|---|---|---|---|---|---|---|---|---|
| | | | Experimental arm (n) | Control arm (n) | | | | | | | | |
| OAHU-COVID19 | NCT04345692 | Inpatient | HCQ (10) | No treatment (6) | 800 mg/day on day 1, then 400 mg/day for 4 days | Confirmed | 18–95 years | 12.50 | United States | None | 350 | Recruiting, not published |
| HYCOVID | NCT04325893 | Inpatient | HCQ (124) | Placebo (123) | 800 mg on day 1, then 400 mg/day for 8 days | Confirmed | ≥18 years | 6.88 | France | Participant, caregiver | 1300 | Terminated, and not published |
| COV-HCQ[f] | NCT04342221 | Inpatient | HCQ (13) | Placebo (14) | 800 mg/day for day 1 and 600 mg/day for days 2–7 | Confirmed | ≥18 years | 3.70 | Germany | Participant, caregiver | 220 | Recruiting, not published |
| COVID-PEP | NCT04308668 | Outpatient | HCQ (244) | Placebo (247) | 800 mg at 0 h, then 600 mg 6–8 h, then 600 mg daily for 4 days | Confirmed and suspected | ≥18 years | 0.41 | International[c] | Participant, caregiver | 3000 | Completed and published |
| BCN PEP CoV-2 Study | NCT04304053 | Outpatient | HCQ (136) | No treatment (157) | 800 mg on day 1, and 400 mg on days 2–7 | Confirmed | ≥18 years | 0 | Spain | None | 2300 | Completed and published |
| | NCT04333654 | Outpatient | HCQ (5) | Placebo (3) | 800 mg at 0 h, then 400 mg 6–8 h later, then 600 mg/day for 9 days | Confirmed | 18–80 years | 0 | International[c] | Participant, caregiver | 210 | Terminated, not published |
| COMIHY | NCT04340544 | Outpatient | HCQ (8) | Placebo (8) | 600 mg/d for 7 days | Confirmed | ≥18 years | 0 | Germany | Participant, caregiver | 2700 | Recruiting, not published |
| **Chloroquine** | | | | | | | | | | | | |
| | ChiCTR2000030054 | Inpatient | CQ (18) | No treatment (12) | 1000 mg/day for 1 day, then 500 mg/day for 9 days | Confirmed | 18–75 years | 0 | China | None | 100 | Completed and published |
| ARCHAIC | NL8490 | Inpatient | CQ (5) | No treatment (3) | 600 mg at 0 h, then 300 mg after 12 h, then 600 mg/day for 4 days | Confirmed | ≥18 years | 12.50 | Netherlands | None | 950 | Terminated, not published |
| CloroCOVID19II | NCT04342650 | Outpatient | CQ (78) | Placebo (74) | 900 mg/day for 1 day, then 450 mg/day for 4 days | Suspected | ≥18 years | 0 | Brazil | Participant, caregiver | 210 | Completed, not published |
| CloroCOVID19III | NCT04323527 | Inpatient | CQ (41) | Placebo (41) | 900 mg/day for 1 day, then 450 mg/day for 4 days | Suspected | ≥18 years | 35.37 | Brazil | Participant, caregiver | 278 | Completed, not published |
| | ChiCTR2000031204 | Inpatient | CQ (18) | Placebo (17) | 1000 mg on day 1, then 500 mg/day on days 2–3, then 250 mg/day until ≤14 days of total treatment | Confirmed | 18 to 70 years | 0 | China | Participant | 300 | Recruiting, not published |

CQ chloroquine, HCQ hydroxychloroquine.
In all trials that used hydroxychloroquine, dosages refer to hydroxychloroquine sulfate. In trials that used chloroquine, the dosages for ARCHAIC, ChiCTR2000030054 and ChiCTR2000031204 refer to chloroquine phosphate, while those for CloroCOVID19II and CloroCOVID19III refer to chloroquine base.
aIncluding peer-reviewed journal publications and posted preprints.
bExcept one study site (University of Pittsburgh Medical Center) that recruited patients in a general ward setting.
cIncluding centers in multiple countries.
dOther arms of the trial are still ongoing.
eTrial includes more treatment arms than reported here; target sample size refers to all arms.
fThe allocation of participants to the experimental arm (n = 13) and control arm (n = 13) was not yet unblinded. The allocation for the meta-analysis was done randomly. A sensitivity analysis with the reverse assignment gave the same result.

**Table 2 Group-level characteristics of included and unavailable trials.**

| | All trials, $n = 62$ | Included trials, $n = 28$ | Potentially eligible, unavailable trials[a], $n = 34$ |
|---|---|---|---|
| **Drug, n (%)** | | | |
| HCQ | 47 (76) | 24 (86) | 23 (68) |
| CQ | 10 (16) | 2 (7) | 8 (22) |
| Both | 5 (8) | 2 (7) | 3 (8) |
| **Planned sample size[a]** | | | |
| Median (IQR) | 355 (150–630) | 500 (218–1350) | 254 (120–442) |
| **Trial status, n (%)** | | | |
| Active, not recruiting | 1 (2) | 0 (0) | 1 (3) |
| Completed | 13 (21) | 12 (43) | 1 (3) |
| Discontinued | 6 (10) | 0 (0) | 6 (18) |
| Not yet recruiting | 2 (3) | 0 (0) | 2 (6) |
| Recruiting | 27 (44) | 8 (29) | 19 (56) |
| Terminated | 13 (21) | 8 (29) | 5 (15) |
| **Location, n (%)** | | | |
| Africa | 3 (5) | 1 (4) | 2 (6) |
| Asia | 23 (37) | 8 (29) | 15 (44) |
| Europe | 16 (26) | 8 (29) | 8 (24) |
| International[b] | 6 (10) | 4 (14) | 2 (6) |
| North America | 10 (16) | 4 (14) | 6 (18) |
| South America | 4 (6) | 3 (11) | 1 (3) |
| **Placebo control, n (%)** | 30 (47) | 11 (39) | 18 (53) |
| **More than two arms, n (%)** | 27 (44) | 10 (37) | 17 (50) |
| **Patient setting, n (%)** | | | |
| ICU | 1 (2) | 1 (4) | 0 |
| Inpatient | 45 (73) | 22 (79) | 23 (68) |
| Outpatient | 12 (19) | 5 (18) | 7 (21) |
| Unclear | 4 (6) | 0 | 4 (12) |
| **Blinding, n (%)** | | | |
| None | 32 (52) | 15 (54) | 17 (50) |
| Outcome assessor | 1 (2) | 1 (4) | 0 |
| Participant | 3 (5) | 1 (4) | 2 (6) |
| Participant, caregiver | 25 (40) | 11 (39) | 14 (41) |
| Participant, outcome assessor | 1 (2) | 0 | 1 (3) |

CQ chloroquine, HCQ hydroxychloroquine, ICU intensive care unit, IQR interquartile range.
[a]Data were extracted from trial registries or publications.
[b]Including centers in multiple countries.

have found no evidence for small-study effects and we consider that the results are unlikely to have been materially affected by publication or reporting bias. This paper offers the most comprehensive summary on HCQ and mortality in COVID-19 to date.

Twenty-three percent of the potentially eligible trials were listed as discontinued, mostly because of fewer patients than expected. Among 28 included RCTs, only two had reached their target sample size at the time of censoring for this meta-analysis. As previously discussed[4], most trials on HCQ and CQ in COVID-19 are small, reflecting both the strong motivation for individual efforts and underscoring the need for readily available research infrastructure to merge small-scale initiatives. Especially in the context of recruitment challenges, we encourage other researchers to form collaborations and combine trial results[35].

Our analysis has some limitations. First, although we adopted a comprehensive, systematic search strategy, our real-time initiative differs from traditional systematic reviews. We focused on collecting unpublished information, aiming to rapidly secure as much trial evidence as possible. We did not review individual trials, nor did we stratify results according to patient characteristics, and we have not collected information on other outcomes than mortality. Such analyses are planned in future publications using in-depth details disclosed in individual trial publications to come[36–38]. The exploratory subgroup analyses did not support the hypothesis that blinding/use of placebo is associated with the observed effect (the test for an interaction gives $p = 0.15$ and the OR is 0.88 with wide CIs 0.55–1.41, compatible with the overall effect); moreover, attrition was negligible (median 0%, IQR 0–0%; range 0–19.5%). A meta-epidemiological study shows little evidence that mortality results would be affected by lack of blinding, or problems in randomization and allocation concealment, in contrast to less objective outcomes[39]. Accordingly, we identified no relevant risk of bias across trials. Second, a majority of the potentially eligible trials were not available. Despite going far beyond the standard review of published evidence, we expect additional results from future trials on CQ to narrow the uncertainty of the treatment effect and possibly reveal benefits or harms not discernible based on the current evidence. We plan to perform an update when substantial additional evidence becomes available. Third, although this analysis intended to combine results from both inpatients and outpatients regardless of disease severity, trials enrolling patients with mild to moderate disease comprised a minority of the final sample size; many of which had zero or few events. Finally, although sensitivity analyses addressing model specifications were compatible with the main analysis, one combination (HKSJ model with SJ $\tau^2$ estimator) yielded substantially wider CIs. This combination gave disproportionately low weight to RECOVERY (15%) and we consider the main model (HKSJ with PM $\tau^2$ estimator) to be more valid in this situation.

Treatment with HCQ for COVID-19 was associated with increased mortality, and there was no benefit from CQ based on currently available randomized trial data. Medical professionals around the globe are encouraged to inform patients about this evidence.

## Methods

This collaborative meta-analysis focused solely on all-cause mortality in order to provide rapid evidence on the most critical clinical outcome. Investigators of ongoing, discontinued, or completed trials were contacted via email to provide group-level (aggregated) mortality data per trial arm at any time point available. The protocol was published online before data collection[40]. This review has been reported in accordance with the Preferred Reporting Items for Systematic Review and Meta-analysis[41]. The PRISMA checklist can be found in the supplement (Supplement 1).

patients. For outpatients who are elderly or have comorbidities, evidence is sparse. Most of the 28 trials excluded persons with comorbid conditions carrying higher risk of adverse events from HCQ or CQ[16,17,27]. No evidence is in the pipeline for these groups, which echoes clinical reasoning being reluctant to expose them to risk.

Although the published trials resulted in a conclusive treatment estimate, the unpublished trials tend towards a null effect. The tendency of published trials to report larger effect sizes than unpublished trials is well-documented and constitute one of the reporting biases that are discernable only when a body of studies are considered together[34]. Null results are less expected to be rapidly disseminated, especially if the trial is small. Of note, RECOVERY results showing dexamethasone benefits have been published more rapidly[3] than the unfavorable HCQ results[13]. We

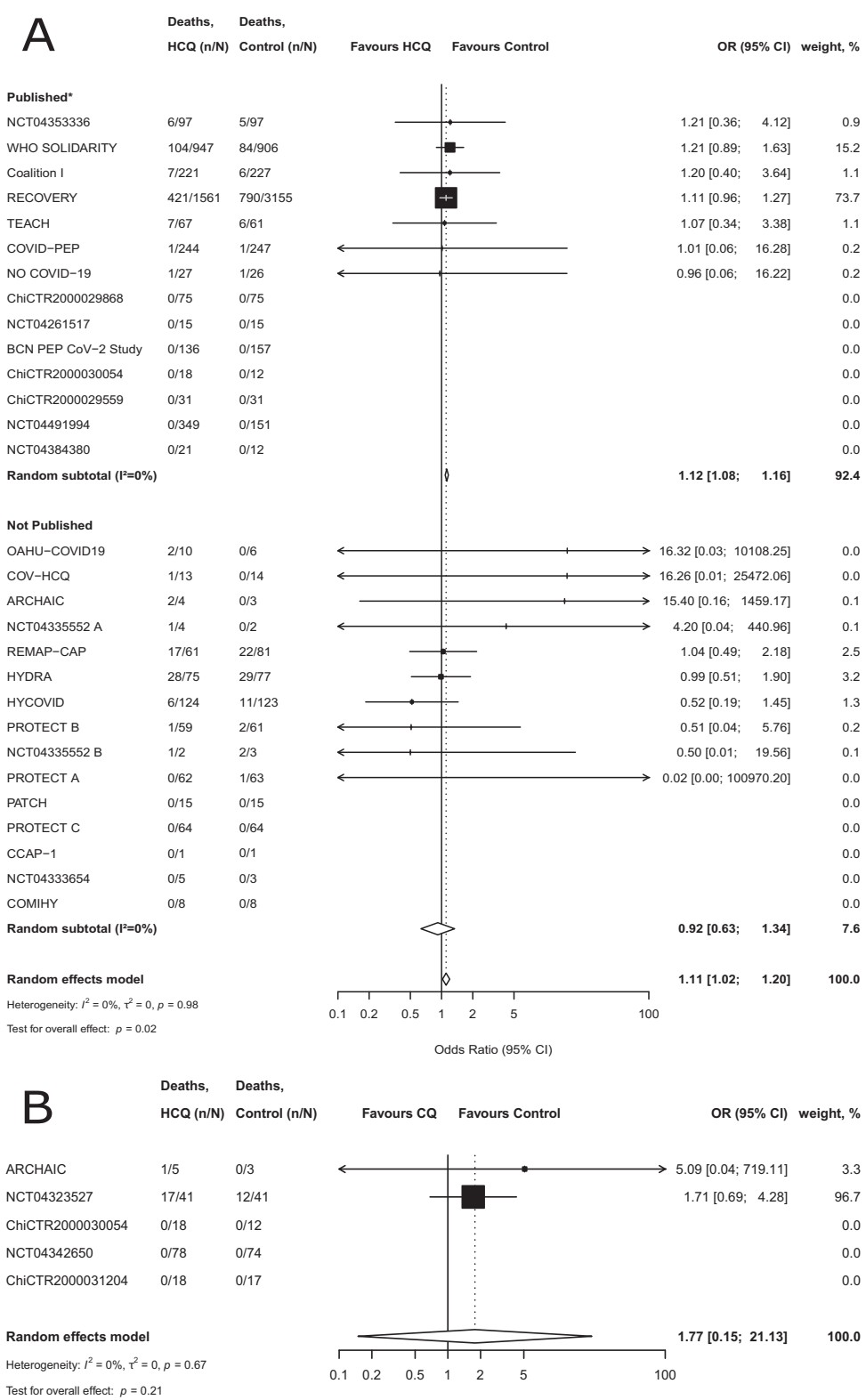

**Eligibility criteria**. We considered all clinical trials that reported randomly allocating patients with confirmed or suspected SARS-CoV-2 infection to a treatment protocol containing HCQ or CQ (for any duration or dose) or the same treatment protocol not containing HCQ or CQ. In other words, the control group had to receive placebo or no treatment other than standard of care (we excluded comparisons of HCQ or CQ against an active treatment, e.g., HCQ versus azithromycin, since active controls were too heterogeneous to pool together and reveal the pure benefits and harms of HCQ or CQ). Eligible ongoing trials had to provide data on all-cause mortality and randomize the first patient before June 1, 2020 (time point selected arbitrarily as we did not expect trials launched later to recruit enough patients to provide relevant additional information). Trials that were published or posted as preprint were not restricted by date. Prevention trials were not included. We included trials regardless of follow-up time and whether mortality was a primary outcome or not; moreover, we put no restrictions on trial status, language, geographical region, or healthcare setting.

**Fig. 2 Random effects meta-analysis for mortality for treatment of COVID-19, trials are stratified by publication status. A** Random-effects meta-analysis for mortality for treatment of COVID-19 with Hydroxychloroquine, trials are stratified by publication status. The dashed vertical line denotes an odds ratio of 1.0, which represents no difference in risk between hydroxychloroquine and the control. The black horizontal bars represent 95% confidence intervals (CIs). Random-effects model of the Hartung–Knapp–Sidik–Jonkman approach was performed to obtain a pooled estimate of the odds ratio. The estimate of heterogeneity ($\tau^2$) was obtained using the Paule and Mandel (PM) estimator. We describe the between-trial heterogeneity using the $I^2$ statistic. The results of the statistical tests for the overall effect and corresponding p values are presented. All tests were two-tailed. *Published as peer-reviewed articles or posted preprints. **B** Random-effects meta-analysis for mortality for treatment of COVID-19 with chloroquine. The dashed vertical line denotes an odds ratio of 1.0, which represents no difference in risk between chloroquine and the control. The black horizontal bars represent 95% confidence intervals (CIs). Random-effects model of the Hartung–Knapp–Sidik–Jonkman approach was performed to obtain a pooled estimate of the odds ratio. The estimate of heterogeneity ($\tau^2$) was obtained using the Paule and Mandel (PM) estimator. We describe the between-trial heterogeneity using the $I^2$ statistic. The results of the statistical tests for the overall effect and corresponding p values are presented. All tests were two-tailed. The x-axis scales differ for reasons of readability. Source data are provided as a Source Data file.

**Search strategy**. We searched for eligible trials registered at ClinicalTrials.gov and the WHO International Clinical Trials Registry Platform (ICTRP) by June 11, 2020 (COVID-evidence database; see Supplement 2)[42]. We additionally searched PubMed and the Cochrane COVID-19 trial registry (covering preprints, trial registries, and literature databases) by June 11, 2020, using terms related to HCQ and CQ combined with terms for COVID-19 and a standard RCT filter (Supplement 3)[43]. We updated the literature search on October 16, 2020. Two authors (C.A. and A.M.S.) independently verified the eligibility criteria (Fig. 1) and solved any discrepancies by discussion.

Principal investigators of 83 potentially eligible trials were asked to confirm the eligibility criteria, as well as: "For each of your study arms: (a) What intervention did this group receive? (b) How many patients were randomized to this group? (c) Of these patients, how many have died? (d) Of these patients, for how many it is unknown if they are dead or alive?" (Supplement 3, email template). Investigators who were not responsive received two email reminders in English or Chinese, depending on trial origin.

**Data extraction**. The following information was extracted from all included RCTs by two reviewers (C.A. and A.M.S.) and verified by the trial investigators: experimental and control arms, number of randomized participants, treatment schedule, patient setting, eligibility criteria, study location, blinding, target sample size, and trial status. In all trials that used hydroxychloroquine, we extracted dosages referring to hydroxychloroquine sulfate. In trials that used chloroquine, the dosages for ARCHAIC, ChiCTR2000030054 and ChiCTR2000031204 refer to chloroquine phosphate, while those for CloroCOVID19II and CloroCOVID19III refer to chloroquine base. We also classified trials as published in a peer-reviewed journal, posted on a preprint server, or unpublished (the latter category not including preprints). For reasons of feasibility within this rapid assessment, we generally did not request descriptive information beyond items included in trial registrations.

**Risk of bias assessment**. Two reviewers (C.A. and A.M.S.) independently assessed the risk of bias of included RCTs using the Cochrane risk of bias tool 2.0[44]. Disagreements were resolved through discussion. We used the information reported in preprints and journal publications, and for unpublished trials, we retrieved information from trial registrations, which was confirmed by trial investigators. We also assessed small-study effects with an inverted funnel plot and Egger's test[43]. The presence of small-study effects may be suggestive, but not definitive, of publication bias[45].

**Data synthesis and analyses**. The main analysis evaluated separately the effect on all-cause mortality of HCQ versus control and CQ versus control. We report absolute numbers and proportions, as well as the treatment effect estimate as an odds ratio (OR; odds of death in the HCQ or CQ intervention group divided by the odds of death in the control group) with 95% CIs. For multi-arm studies, we requested data for all arms and calculated treatment effect estimates for each eligible comparison. We combined mortality effects from all RCTs based on binary outcome data (2 × 2 contingency tables) in meta-analyses and describe the statistical heterogeneity using the $I^2$ statistic[46]. In our protocol, we prespecified a random-effects model of the Hartung–Knapp–Sidik–Jonkman (HKSJ) approach[47], in order to provide more equality of weights between trials with moderate to large size (than, e.g., the DerSimonian–Laird approach). We did not prespecify the between-study variance estimator, $\tau^2$, but chose the Paule and Mandel (PM) estimator based on provided guidance on choosing among 16 variants[48]. Cases of zero events in one arm were corrected by adding the reciprocal of the size of the contrasting study arm[43]. However, considering the range of sample sizes and the number of zero events across trials, we assessed the effects of alternative approaches with sensitivity analyses, as detailed below. To explore and illustrate evidence generation over time, we also performed a cumulative meta-analysis of all trials as well as stratified by dissemination status (publications/preprints versus unpublished), using the HKSJ approach with PM $\tau^2$. We used the date of email response or publication/posting of preprint. The meta-analyses were completed using R version 3.5.1 and the "meta" package version 4.13-0.

**Subgroup analyses**. In exploratory subgroup analyses, we stratified trials by patient setting (as proxy to COVID-19 severity: outpatients, inpatients but not intensive care unit (ICU), and ICU), diagnostic confirmation (confirmed SARS-CoV-2 versus suspected cases), control type (placebo control versus other), and publications/preprints versus unpublished trials. We did not stratify for missing data since the amount was extremely low. A post hoc stratification by HCQ dose was added (trials with ≥1600 mg on day 1 and ≥800 mg from day 2 versus lower-dose trials) to isolate trials predicted to achieve blood levels of HCQ above the in vitro half-maximal inhibitory concentration value for SARS-CoV-2 (1.13 μM)[49].

**Sensitivity analyses**. We added exploratory sensitivity analyses to assess robustness across meta-analytic approaches: DerSimonian–Laird and Sidik–Jonkman $\tau^2$ estimators, Mantel–Haenszel random-effects method, and Peto method. DerSimonian–Laird is a standard random-effects meta-analysis approach, but may underestimate uncertainty. The Sidik–Jonkman $\tau^2$ estimator, on the other hand, may yield inflated estimates if heterogeneity is low[48]. The Mantel–Haenszel method performs reasonably well with small and zero event counts, much like Peto and arcsine transformation. The Peto method is suboptimal in the presence of substantial imbalances in the allocation ratio of patients randomized in the compared arms (e.g., RECOVERY trial). We also modeled variants to handling zero events (arcsine difference, and excluding trials with zero events) as well as excluding trials with <50 participants.

**Unpublished trial details**. All unpublished trials were performed according to the principles of the Declaration of Helsinki and written informed consent was obtained from the study participants. The release of mortality outcome data was authorized by the respective data and safety monitoring boards and principal investigators. Ethical approval was granted by institutional review boards as follows: University of Pennsylvania, ref. #842838 (PATCH, NCT04329923); National Bioethics Committee (NBC) Pakistan, ref. 4–87/NBC-471-COVID-19-05/20/ (PROTECT, NCT04338698); Ethics Committee of the Capital Region of Denmark, ref. H-20025317 (CCAP-1, NCT04345289); Comité de Protection des Personnes du Sud-Ouest et Outre-Mer 4, ref. CPP2020-03-036/2020-001271-33/ 20.03.24.72431, and the Agence Nationale de Sécurité du Médicament et des produits de santé (ANSM), ref. MEDAECNAT-2020-03-00045 (HYCOVID, NCT04325893); Ethik-Kommission an der Medizinischen Fakultät der Eberhard-Karls-Universität und am Universitätsklinikum Tübingen, ref. 190/2020AMG1 and ref. 225/2020AMG1 (COV-HCQ, NCT04342221; and COMIHY, NCT04340544, respectively); London-Surrey Borders Research Ethics Committee in the UK, Medisch Ethische Toetsingscommissie Utrecht (METC Utrecht) in the Netherlands, Sydney Local District ethics Review Committee (Royal Prince Alfred Hospital) in Australia, Northern A Health and Disability Ethics Committee in New Zealand, St. Vincent's Healthcare Group Ethics and Medical Research Committee in Ireland, King Abdullah International Medical Research Center Institutional Review Board in Saudi Arabia, University of Pittsburgh Institutional Review Board in the United States, Unity Health Research Ethics Board in Canada, National Ethics Committee for Clinical Research (CEIC) in Portugal, and the Romania Academy of Medical Sciences National Bioethics Committee for Medicines and Medical Devices (REMAP-CAP, NCT02735707); Comissão Nacional de Ética em Pesquisa (CONEP), ref. 3.961.681 (CloroCOVID19II A, NCT04323527, and CloroCOVID19II B, NCT04342650); a Single Ethics Committee from the Coordination of the National Institutes of Health and High Specialty Hospitals, ref. C13-20 (HYDRA, NCT04315896); the Ethics Committee of Beijing Youan Hospital, Capital Medical University, ref. JINYOUKELUN[2020]013 (ChiCTR2000031204); Partners Human Research Committee at the Brigham and Women's Hospital, Boston, Ethisch Comite in Belgium, and Stichting Beoordeling Ethiek Biomedisch in the Netherlands (NCT04333654); The Queen's Medical Center, ref. RA-2020-018 (OAHU-COVID19, NCT04345692); Duke University Medical Center Institutional Review Board, ref. Pro00105339, and UnityPoint Health Institutional Review Board (NCT04335552); and Medical Ethics Committee Utrecht (METC Utrecht), part of the Dutch Central Committee on Research Involving Human Subjects (ARCHAIC, NL8490).

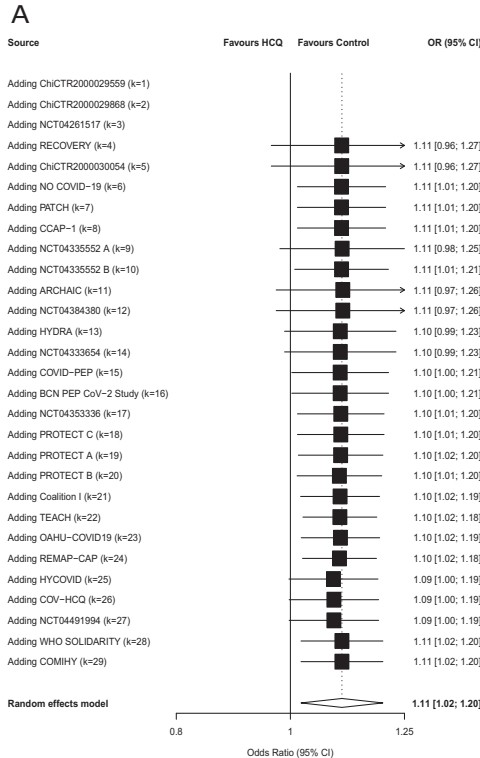

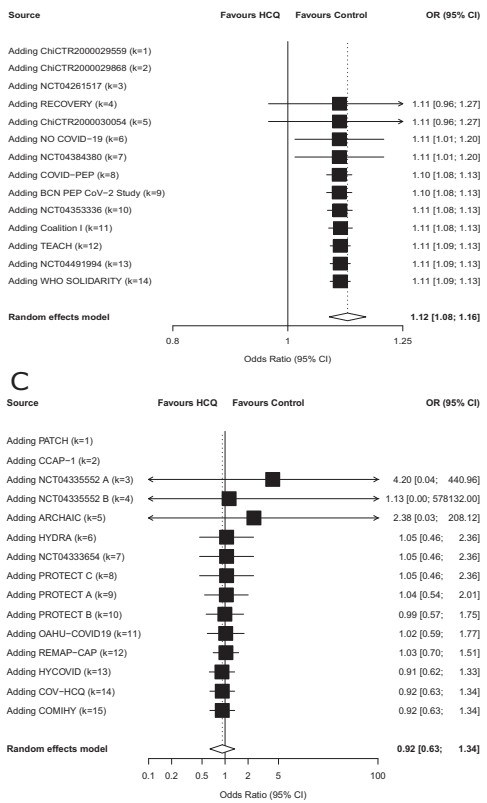

**Fig. 3 Cumulative meta-analysis for mortality for treatment of COVID-19 with Hydroxychloroquine. A** Cumulative meta-analysis for mortality for treatment of COVID-19 with hydroxychloroquine. HCQ was evaluated in 26 trials with 10,012 patients. Four thousand three hundred and sixteen patients were treated with hydroxychloroquine, of whom 606 died. Five thousand six hundred and ninety-six patients were allocated to the control group, of whom 960 died. The dashed vertical line denotes an odds ratio of 1.0, which represents no difference in risk between hydroxychloroquine and the control. The black horizontal bars represent 95% confidence intervals (CIs). Random-effects model of the Hartung–Knapp–Sidik–Jonkman approach was performed to obtain a pooled estimate of the odds ratio. The estimate of heterogeneity ($\tau^2$) was obtained using the Paule and Mandel (PM) estimator. We describe the between-trial heterogeneity using the $I^2$ statistic. The results of the statistical tests for the overall effect and corresponding $p$ values are presented. All tests were two-tailed. **B** Cumulative meta-analysis for mortality for treatment of COVID-19 with hydroxychloroquine (publications and preprints only). HCQ was evaluated in 14 published trials with 8981 patients. Three thousand eight hundred and nine patients were treated with hydroxychloroquine, of whom 547 died. Five thousand one hundred and seventy-two patients were allocated to the control group, of whom 893 died. The dashed vertical line denotes an odds ratio of 1.0, which represents no difference in risk between hydroxychloroquine and the control. The black horizontal bars represent 95% confidence intervals (CI). Random-effects model of the Hartung–Knapp–Sidik–Jonkman approach was performed to obtain a pooled estimate of the odds ratio. The estimate of heterogeneity ($\tau^2$) was obtained using the Paule and Mandel (PM) estimator. We describe the between-trial heterogeneity using the $I^2$ statistic. The results of the statistical tests for the overall effect and corresponding $p$ values are presented. All tests were two-tailed. **C** Cumulative meta-analysis for mortality for treatment of COVID-19 with hydroxychloroquine (unpublished data only). HCQ was evaluated in 12 unpublished trials with 1031 patients. Five hundred and seven patients were treated with hydroxychloroquine, of whom 59 died. Five hundred and twenty-four patients were allocated to the control group, of whom 67 died. The dashed vertical line denotes an odds ratio of 1.0, which represents no difference in risk between hydroxychloroquine and the control. The black horizontal bars represent 95% confidence intervals (CIs). Random-effects model of the Hartung–Knapp–Sidik–Jonkman approach was performed to obtain a pooled estimate of the odds ratio. The estimate of heterogeneity ($\tau^2$) was obtained using the Paule and Mandel (PM) estimator. We describe the between-trial heterogeneity using the $I^2$ statistic. The results of the statistical tests for the overall effect and corresponding $p$ values are presented. All tests were two-tailed. The $x$-axis scales differ for reasons of readability. Source data are provided as a Source Data file.

## Code availability

The code file is provided in the Open Science Framework [https://osf.io/qesv4/][40].

**Reporting summary**. Further information on research design is available in the Nature Research Reporting Summary linked to this article.

## Data availability

All trial-level data generated or analyzed during this study are included in this published article and its Supplementary information files. The data file is provided in the Open Science Framework [https://osf.io/qesv4/][40]. Source data are provided with this paper.

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

## Acknowledgements

We wish to express our heartfelt gratitude to all patients volunteering for the trials involved. We furthermore thank Wenyan Ma and Benjamin Kasenda (University Hospital Basel, University of Basel) for kindly translating the emails to Chinese investigators. For valuable contributions to individual trials in this collaborative group, we sincerely thank: Hannah Jin, Monica Feeley, Bruce Bausk, Jessica Cauley, Jane Kleinjan, Jon Gothing, Naeema Bangash, Heather Wroe, Claire Bigogne, Christelle Castell, Annelies Mottart, Lisette Cortenraad, Judith Medema-Muller, Katia Handelberg, Khalid Ben-hammou, Shaheen Kumar, Sophie Gribomont, Kim Kuehne, Cathia Markina, Julien Labeirie, Julie Pencole, Eva Crispyn, Cecile Le Breton, Kelli Horn, Tina Patel, Benjamin Harrois, Isabelle Collin, Vetheeswar Manivannan, Irma Slomp, Frederick Becue, Isabelle Godefroy, Lynne Guo, Lene Kollmorgen, Toluwalope Cole, Catherine Chene, Praveena Deenumsetti, Anne Doisy, Ariane Vialfont, Melissa Charbit, Christine Shu, Stephane Kirkesseli, Howard Surks, Magalie De Meyer, Edel Hendrickx, and Paul Deutsch (Sanofi trial); Ellie Carmody, Märtin Backer, Jaishvi Eapen, Jack A. DeHovitz, Prithiv J. Prasad, Yi Li, Camila Delgado, Morris Jrada, Gabriel A. Robbins, Brooklyn Henderson,

Alexander Hrycko, Dinuli Delpachitra, Vanessa Raabe, Jonathan S. Austrian, and Yanina Dubrovskaya (TEACH trial); Farah Al-Beidh, Djillali Annane, Kenneth Baillie, Abigail Beane, Richard Beasley, Zahra Bhimani, Marc Bonten, Charlotte Bradbury, Frank Brunkhorst, Meredith Buxton, Allen Cheng, Menno de Jong, Eamon Duffy, Lise Estcourt, Rob Fowler, Timothy Girard, Herman Goossens, Cameron Green, Rashan Haniffa, Christopher Horvat, David Huang, Francois Lamontagne, Patrick Lawler, Kelsey Linstrum, Edward Litton, John Marshall, Daniel McAuley, Shay McGuinness, Stephanie Montgomery, Paul Mouncey, Katrina Orr, Rachael Parke, Jane Parker, Asad Patanwala, Kathryn Rowan, Marlene Santos, Christopher Seymour, Steven Tong, Anne Turner, Timothy Uyeki, Wilma van Bentum-Puijk, Frank van de Veerdonk, and Ryan Zarychanski (REMAP-CAP trial); Jan-Erik Berdal, Arne Eskesen, Dag Kvale, Inge Christoffer Olsen, Corina Silvia Rueegg, Anbjørg Rangberg, Christine Monceyron Jonassen, and Torbjørn Omland (NO COVID-19 trial). Support for title page creation and format was provided by AuthorArranger, a tool developed at the National Cancer Institute.

This collaborative meta-analysis was supported by the Swiss National Science Foundation and Laura and John Arnold Foundation (grant supporting the postdoctoral fellowship at the Meta-Research Innovation Center at Stanford (METRICS), Stanford University). Funding also includes postdoctoral grants from Uppsala University, the Swedish Society of Medicine, the Blanceflor Foundation, and the Sweden-America Foundation (C.A.). The funders had no role in the design of this collaborative meta-analysis; in the collection, analysis, and interpretation of data; or in the report writing. The corresponding author had full access to all study data and final responsibility for the decision to submit for publication.

## Author contributions

L.G.H., C.A., and A.M.S. had full access to all data in this study and take responsibility for the integrity of the data and the accuracy of the data analysis. Concept and design: L.G.H., J.P.A.I., C.A., A.M.S., S.N.G., and D.M. Acquisition, analysis, or interpretation of data: all authors. Drafting of the manuscript: C.A., A.M.S., and L.G.H. Critical revision of the manuscript for important intellectual content: all authors. Statistical analysis: A.M.S., L.G.H., J.P.A.I., S.N.G., and P.J. Approval of the final manuscript: all authors. Obtained funding: L.G.H., C.A., and J.P.A.I. Administrative, technical, or material support: C.A., A.M.S., L.G.H., P.J., and J.v.H. Supervision: L.G.H.

## Competing interests

B.S.A. and R.K.A. are the primary investigators of the Prevention and Treatment of COVID19 with Hydroxychloroquine (PATCH) trial, funded by a philanthropic gift. R.K.A reports being founder with equity of Pinpoint Therapeutics and Immunacell, and personal fees from Sprint Biosciences and Dec.iphera, outside the submitted work. D.C.A. reports personal fees from Ferring Pharmaceuticals, Inc., Bristol-Myers Squibb, and Bayer AG, other from Alung Technologies, Inc., outside the submitted work; in addition, D.C.A. has pending patents for Selepressin—compounds, compositions, and methods for treating sepsis to Ferring, B.V., and Proteomic biomarkers of sepsis in elderly patients pending to University of Pittsburgh. Y.M.A. reports that he is the principal investigator on a clinical trial of lopinavir–ritonavir and interferon for Middle East respiratory syndrome (MERS) and that he was a nonpaid consultant on therapeutics for MERS-coronavirus (CoV) for Gilead Sciences and SAB Biotheraputics. He is a coinvestigator on the Randomized, Embedded, Multi-factorial Adaptive Platform Trial for Community-Acquired Pneumonia (REMAP-CAP), a board member of the International Severe Acute Respiratory and Emerging Infection Consortium (ISARIC), and the Lead-Co Chair of the Think20 (T20) Taskforce for COVID-19. Brigham and Women's Hospital, PRA Health Science, and Cliniques universitaires Saint-Luc received funds from Sanofi. T.B. reports grants from Pfizer, Novo Nordisk Foundation, Simonsen Foundation, Lundbeck Foundation, and Kai Hansen Foundation; grants and personal fees from GSK, Pfizer, Boehringer Ingelheim, and Gilead; and personal fees from MSD, all outside the submitted work. Y.Z.C., L.N.C., B.I., and L.P. are employees of Sanofi. The COV-HCQ and COMIHY trials were supported by the German Federal Ministry of Education and Research (EudraCT number 2020-001224-33) and the German Federal Ministry of Health (EudraCT number 2020-001512-26). L.D. reports grants from EU FP7-HEALTH-2013-INNOVATION-1, grant number 602525, grants from H2020 RECOVER, grant agreement no. 101003589, during the conduct of the study; and is a member of the COVID-19 guideline committee SCCM/ESICM/SSC, member of the ESICM COVID-19 taskforce, and chair of the Dutch intensivists (NVIC) taskforce infectious threats. V.D.

reports nonfinancial support from MSD France and from Sanofi Aventis France, outside the submitted work. A.E. is an employee of Ividata Life Sciences and works as an external contractor for Sanofi. A.C.G. received grant funding from an NIHR Research Professorship (RP-2015-06-18), support from the NIHR Imperial Biomedical Research Centre, and consulting fees paid to his institution from GlaxoSmithKline and Bristol-Myers Squibb. T.H. reports grants from the Health Research Council of New Zealand, during the conduct of the study. A.I.M.H. reports grants from ZonMw, Netherlands organisation for Health Research and Development, during the conduct of the study. HYDRA trial was an investigator-initiated study supported by Sanofi, CONACYT (National Council of Science and Technology of Mexico) and by the participating centers. Thuy Le reports grants from Gilead Sciences, outside the submitted work. B.J.M. reports grants from NIH/NHLBI, and from Bayer Pharmaceuticals, Inc., outside the submitted work. S.M. receives funding as the Innovative Medicines Canada Chair in Pandemic Preparedness. C.M. reports grants from the Health Research Council of New Zealand. M.J.M. reports having received the HCQ drug from the New York State government, during the conduct of the study; grants from Lilly, Pfizer, Sanofi, and personal fees from Meissa, outside the submitted work; in addition, M.J.M. has a patent anti-Zika monoclonal ab/ Emory Univ pending. A.N. is supported by a Health Research Board of Ireland Clinical Trial Network Award (HRB-CTH-2014-012). L.P. is an employee of Excelya and works as an external contractor for Sanofi. F.W.R. reports personal fees from Merck Research Labs, Novartis, Lilly, Sanofi, NovoNordisk, KLSMC, Tolerion, Rhythm, UCB, AstraZeneca, Janssen, Merck KGaA, Sarepta, Eidos, Amgen, Phathom, outside the submitted work; and having equity interest in GlaxoSmithkline, Athira Pharma, Data-Vant, Spencer Healthcare. S.R.W. reports receiving a grant from Sanofi during the conduct of the study and grants from NIH-NIAID outside the submitted work, and having conducted vaccine (HIV, Zika) clinical trials funded by Janssen. S.A.W. reports grants from National Health and Medical Research Council (Australia), grants from Minderoo Foundation, from Health Research Council (New Zealand), and from Medical Research Future Fund (Australia), during the conduct of the study. J.M.W. reports grants from ZonMw, Netherlands organisation for Health Research and Development, during the conduct of the study. F.G.Z. was part of the Coalition 1 trial partially supported by EMS Pharmaceuticals, has received previous grants from Bactiguard, Sweden, outside the submitted work, and support from Baxter LA for another clinical trial in critically ill patients. None of the other authors have any competing interests to declare.

## Additional information

Cathrine Axfors[1,2,86], Andreas M. Schmitt [3,4,86], Perrine Janiaud[3], Janneke van't Hooft [1,5], Sherief Abd-Elsalam [6], Ehab F. Abdo [7], Benjamin S. Abella [8], Javed Akram[9], Ravi K. Amaravadi[10], Derek C. Angus[11,12], Yaseen M. Arabi[13], Shehnoor Azhar[14], Lindsey R. Baden[15], Arthur W. Baker [16],

Leila Belkhir[17], Thomas Benfield[18], Marvin A. H. Berrevoets[19], Cheng-Pin Chen[20], Tsung-Chia Chen[21], Shu-Hsing Cheng[20], Chien-Yu Cheng[20], Wei-Sheng Chung[21], Yehuda Z. Cohen[22], Lisa N. Cowan[22], Olav Dalgard[23,24], Fernando F. de Almeida e Val[25], Marcus V. G. de Lacerda[25,26], Gisely C. de Melo[25,27], Lennie Derde[28,29], Vincent Dubee[30], Anissa Elfakir[31], Anthony C. Gordon[32], Carmen M. Hernandez-Cardenas[33], Thomas Hills[34,35], Andy I. M. Hoepelman[36], Yi-Wen Huang[37], Bruno Igau[22], Ronghua Jin[38], Felipe Jurado-Camacho[33], Khalid S. Khan[39], Peter G. Kremsner[40,41,42], Benno Kreuels[43,44], Cheng-Yu Kuo[45], Thuy Le[16], Yi-Chun Lin[20], Wu-Pu Lin[46], Tse-Hung Lin[37], Magnus Nakrem Lyngbakken[24,47], Colin McArthur[34,35,48], Bryan J. McVerry[49], Patricia Meza-Meneses[50], Wuelton M. Monteiro[25,27], Susan C. Morpeth[51], Ahmad Mourad[52], Mark J. Mulligan[53,54], Srinivas Murthy[55], Susanna Naggie[16], Shanti Narayanasamy[16], Alistair Nichol[48,56,57,58], Lewis A. Novack[59], Sean M. O'Brien[60], Nwora Lance Okeke[16], Léna Perez[61], Rogelio Perez-Padilla[62], Laurent Perrin[63], Arantxa Remigio-Luna[62], Norma E. Rivera-Martinez[64], Frank W. Rockhold[60], Sebastian Rodriguez-Llamazares[62], Robert Rolfe[16], Rossana Rosa[65], Helge Røsjø[24,66], Vanderson S. Sampaio[25,67], Todd B. Seto[68,69], Muhammad Shahzad[70], Shaimaa Soliman[71], Jason E. Stout[16], Ireri Thirion-Romero[62], Andrea B. Troxel[72], Ting-Yu Tseng[21], Nicholas A. Turner[16], Robert J. Ulrich[73], Stephen R. Walsh[15], Steve A. Webb[48,74], Jesper M. Weehuizen[36], Maria Velinova[75], Hon-Lai Wong[76], Rebekah Wrenn[16], Fernando G. Zampieri[77,78,79], Wu Zhong[80], David Moher[81], Steven N. Goodman[1,82,83], John P. A. Ioannidis[1,82,83,84,85] & Lars G. Hemkens[1,3,85] ✉

[1]Meta-Research Innovation Center at Stanford (METRICS), Stanford University, Stanford, CA, USA. [2]Department for Women's and Children's Health, Uppsala University, Uppsala, Sweden. [3]Department of Clinical Research, University Hospital Basel, University of Basel, Basel, Switzerland. [4]Department of Medical Oncology, University of Basel, Basel, Switzerland. [5]Amsterdam University Medical Center, Amsterdam University, Amsterdam, the Netherlands. [6]Tropical Medicine and Infectious Diseases Department, Faculty of Medicine, Tanta University, Tanta, Egypt. [7]Tropical Medicine and Gastroenterology Department, Faculty of Medicine, Assiut University, Assiut, Egypt. [8]Department of Emergency Medicine, University of Pennsylvania, Philadelphia, PA, USA. [9]Department of Internal Medicine, Vice Chancellor, University of Health Sciences, Lahore, Punjab, Pakistan. [10]Abramson Cancer Center and Department of Medicine, University of Pennsylvania, Philadelphia, PA, USA. [11]Department of Critical Care Medicine, The Clinical Research Investigation and Systems Modeling of Acute Illness (CRISMA) Center, University of Pittsburgh, Pittsburgh, PA, USA. [12]the UPMC Health System Office of Healthcare Innovation, University of Pittsburgh Medical Centre, Pittsburgh, PA, USA. [13]Intensive Care Department, King Saud Bin Abdulaziz University for Health Sciences and King Abdullah International Medical Research Center, Riyadh, Saudi Arabia. [14]Department of Public Health, University of Health Sciences, Lahore, Punjab, Pakistan. [15]Division of Infectious Diseases, Brigham and Women's Hospital, Boston, MA, USA. [16]Department of Medicine, Division of Infectious Diseases and International Health, Duke University Medical Center, Durham, NC, USA. [17]Infectious Diseases Department, Cliniques universitaires Saint-Luc, Université Catholique de Louvain, Brussels, Belgium. [18]Center of Research & Disruption of Infectious Diseases, Department of Infectious Diseases, Copenhagen University Hospital, Amager and Hvidovre, Hvidovre, Denmark. [19]Department of Internal Medicine, Elisabeth-Tweesteden hospital, Tilburg, Netherlands. [20]Department of Infectious Diseases, Taoyuan General Hospital, Ministry of Health and Welfare, Taoyuan, Taiwan. [21]Department of Internal Medicine, Taichung Hospital, Ministry of Health and Welfare, Taichung, Taiwan. [22]Sanofi, Bridgewater, NJ, USA. [23]Department of Infectious Diseases, Division of Medicine, Akershus University Hospital, Lørenskog, Norway. [24]Institute of Clinical Medicine, Faculty of Medicine, University of Oslo, Oslo, Norway. [25]Fundação de Medicina Tropical Dr. Heitor Vieira Dourado, Manaus, AM, Brazil. [26]Instituto Leonidas e Maria Deane – ILMD, FIOCRUZ-AM, Manaus, AM, Brazil. [27]Universidade do Estado do Amazonas, Manaus, AM, Brazil. [28]Julius Center for Health Sciences and Primary Care, University Medical Center Utrecht, Utrecht, Netherlands. [29]Intensive Care Centre, University Medical Center Utrecht, Utrecht, Netherlands. [30]Infectious and Tropical Diseases Department, Angers University Hospital, Angers, France. [31]Ividata Life Sciences, Levallois-Perret, France. [32]Department of Surgery and Cancer, Anaesthetics, Pain Medicine, and Intensive Care Medicine, Imperial College London and Imperial College Healthcare NHS Trust, London, UK. [33]Critical Care Department, Instituto Nacional de Enfermedades Respiratorias Ismael Cosío Villegas, Ciudad de México, Mexico. [34]Medical Research Institute of New Zealand, Wellington, New Zealand. [35]Auckland City Hospital, Auckland, New Zealand. [36]Department of Infectious Diseases, University Medical Center Utrecht, Utrecht, Netherlands. [37]Department of Internal Medicine, Chang Hua Hospital, Ministry of Health and Welfare, Changhua, Taiwan. [38]Beijing Youan Hospital, Capital Medical University, Beijing, People's Republic of China. [39]Department of Preventive Medicine & Public Health, University of Granada, Hospital Real, Avenida del Hospicio, Granada, Granada, Spain. [40]Institute of Tropical Medicine, University of Tübingen, Tübingen, Germany. [41]Centre de Recherches Médicales de Lambaréné, Lambaréné, Gabon. [42]German Center for Infection Research, Partner Site Tübingen, Tübingen, Germany. [43]Department of Medicine, Division of Tropical Medicine and Division of Infectious Diseases, University Medical Center Hamburg-Eppendorf, Hamburg, Germany. [44]Department of Tropical Medicine, Bernhard Nocht Institute for Tropical Medicine, Hamburg, Germany. [45]Department of Internal Medicine, Pingtung Hospital, Ministry of Health and Welfare, Pingtung, Taiwan. [46]Department of Internal Medicine, Taipei Hospital, Ministry of Health and Welfare, New Taipei City, Taiwan. [47]Division of Medicine, Akershus University Hospital, Lørenskog, Norway. [48]School of Epidemiology and Preventive Medicine, Australian and New Zealand Intensive Care Research Centre, Monash University, Melbourne, VIC, Australia. [49]Department of Medicine, University of Pittsburgh, Pittsburgh, PA, USA. [50]Hospital Regional de Alta especialidad de Ixtapaluca, Ixtapaluca, Mexico. [51]Middlemore Hospital, Auckland, New Zealand. [52]Department of Medicine, Duke University Medical Center, Durham, NC 27710, USA. [53]Department of Microbiology, NYU Grossman School of Medicine, New York, NY, USA. [54]Department of Internal Medicine, Division of Infectious Diseases and Immunology, NYU Grossman School of Medicine, New York, NY, USA. [55]University of British Columbia School of Medicine, Vancouver, BC, Canada. [56]Department of

Intensive Care, Alfred Health, Melbourne, VIC, Australia. [57]Department of Anesthesia and Intensive Care, St Vincent's University Hospital, Dublin, Ireland. [58]School of Medicine and Medical Sciences, University College Dublin, Dublin, Ireland. [59]Division of Infectious Diseases, Brigham and Women's Hospital, Harvard Medical School, Boston, MA, USA. [60]Department of Biostatistics and Bioinformatics, Duke University Medical Center and Duke Clinical Research Institute, Durham, NC, USA. [61]Excelya, Montpellier, France. [62]Department of Smoking and COPD, Instituto Nacional de Enfermedades Respiratorias Ismael Cosío Villegas, Ciudad de México, Mexico. [63]Sanofi, Montpellier, France. [64]Hospital Regional de Alta especialidad de Oaxaca, Oaxaca, Mexico. [65]UnityPoint Health, Des Moines, IA, USA. [66]Division of Research and Innovation, Akershus University Hospital, Lørenskog, Norway. [67]Fundação de Vigilância em Saúde do Amazonas, Manaus, AM, Brazil. [68]University of Hawaii John A. Burns School of Medicine, Honolulu, HI, USA. [69]The Queen's Medical Center, Honolulu, HI, USA. [70]Department of Pharmacology, University of Health Sciences, Lahore, Punjab, Pakistan. [71]Public Health and Community Medicine, Menoufia University, Menoufia, Egypt. [72]Division of Biostatistics, Department of Population Health, NYU Grossman School of Medicine, New York, NY, USA. [73]Department of Medicine, Division of Infectious Diseases and Immunology, NYU Grossman School of Medicine, New York, NY, USA. [74]St. John of God Hospital, Subiaco, WA, Australia. [75]PRA Health Science, Groningen, Netherlands. [76]Department of Internal Medicine, Keelung Hospital, Ministry of Health and Welfare, Keelung, Taiwan. [77]Research Institute, HCor-Hospital do Coração, São Paulo, Brazil. [78]Research Institute, BRICNet - Brazilian Research in Intensive Care Network, São Paulo, Brazil. [79]IDor Research Institute, São Paulo, Brazil. [80]National Engineering Research Center for the Emergency Drug, Beijing Institute of Pharmacology and Toxicology, Beijing, People's Republic of China. [81]Centre for Journalology, Clinical Epidemiology Program, Ottawa Hospital Research Institute, Ottawa, ON, Canada. [82]Stanford University School of Medicine, Stanford, CA, USA. [83]Department of Epidemiology and Population Health, Stanford University School of Medicine, Stanford, CA, USA. [84]Stanford Prevention Research Center, Department of Medicine, Stanford University, Stanford, CA, USA. [85]Meta-Research Innovation Center Berlin (METRIC-B), Berlin Institute of Health, Berlin, Germany. [86]These authors contributed equally: Cathrine Axfors, Andreas M. Schmitt. ✉email: lars.hemkens@usb.ch

