## [Peer Review File · Nature Communications]

Reviewers' Comments:

Reviewer #1:

Remarks to the Author:

Axfors and colleagues conducted a systematic review and meta-analysis on the effect of HCQ/CQ in treatment of COVID-19, which is a very important scientific question to be answered in searching for treatment options during the COVID-19 pandemic. The authors identified 28 trials (with 10319 patients) after thorough literature retrieval and contacting authors for additional necessary information. The manuscript was updated to include the latest WHO SOLIDARITY results. They concluded that HCQ/CQ did not have benefit on the survival of COVID-19 patients, and HCQ was associated with increased mortality.

Although there have been a few other systematic reviews with similar findings, this manuscript provided updated results and focused solely on RCTs. However, there are several points for further consideration:

1. My major concern is that about half of the included studies (12 studies, 1282 patients) had zero deaths and were therefore actually "excluded" (given zero weight) in the meta-analysis. There were also many studies with very few event numbers and were given limited weight in data combination. These studies often focus on mild/moderate COVID-19 patients or outpatients, whereas the RECOVERY trial (with 73.7% weight in meta-analysis) included a large number of severe hospitalized patients. Although this review intended to combine results from both inpatients and outpatients, as well patients of various disease severity, it actually shifts towards severe hospitalized COVID-19 patients when all-cause mortality was the only outcome of interest. This limitation shall be carefully addressed in the manuscript.
2. Although mortality is the only outcome focused in the review, I would suggest also briefly discussing potential effects of HCQ in other outcomes, e.g. possible reduction in viral load, especially when many included studies did not have mortality data. Alternatively, a narrative summary for additional outcomes may be performed for those studies with zero deaths.
3. Studies allocating patients with confirmed or suspected SARS-CoV-2 infection were included. It would be nice if studies with suspected SARS-CoV-2 were marked in Table1.
4. There was no assessment for publication bias of the included studies. Funnel plots or Egger's test should be performed.

Reviewer #2:

Remarks to the Author:

Firstly I'd like to commend the large collaborative effort of the authors to go beyond the existing published data in an attempt to synthesise the effects of HCQ and CQ on mortality associated with COVID-19. Unlike other meta-analyses, they have been able to include a considerable number of additional, unpublished trials, albeit that these only contribute only a relatively small amount (weight) of information. I don't think the authors would have been able to anticipate this.

Major

The statistical methods seem robust, and the conclusions may well be valid, the authors discuss the issue of reporting bias in relation to the included trials, but do not discuss the potential impact of the unavailable trials on their results. While those that are still recruiting may struggle to do so now, those that have completed or been discontinued should have results that could contribute. It strikes me that they may not have heard from some investigators because their trials had negative or null results, in which case this might be an optimistic view of the effects of HCQ/CQ.

Another omission is any assessment of the 'quality' or risk of bias of the included trials. As stated in the protocol, all cause mortality may limit detection biases and they describe blinding in the

text, but what about the potential for other biases? This is important particularly for unpublished trials and data, which at the moment need to be taken at face value. I understand that time is of the essence, but even a basic assessment of e.g. , the randomisation process, would re-assure the reader as to the quality of the trials, and evidence presented. This should be straightforward for published trials and presumably, for unpublished trials, could be done on the basis of trial registration data together with investigator input. I don't think it is valid to postpone this a future publication.

Was all-cause mortality measured at same time point e.g. 28 days in all trials? If not, could it have a bearing on the results?

I find the main manuscript text, tables and the figures in particular to give too much of a meta-analysis methodology perspective (e.g. publication status, comparing models, cumulative meta-analysis). Although these add to the interpretation and should be referred to, it would be better if the data presented was geared to providing clinical insight? For example, I would suggest grouping table 1 by drug, then setting, and control group to be more important than published vs unpublished. Also, it would be good to include at the very least HCQ subgroup plots by setting and control comparator as well as the published versus not in the main manuscript. For these, it is important to see the data (events/pts) on the plot (not just table S2A). Given that a few trials contribute most of the the cumulative meta-analysis plots are less informative, and so could be relegated to the appendix.

Minor

The discussion could be better structured better to make the messages clearer, for example: key findings, context, strength, limitations and implications.

Mortality outcomes with hydroxychloroquine and chloroquine in COVID-19: an international collaborative meta-analysis of randomized trials

NCOMMS-20-39123

Point by point reply

EDITORIAL REQUIREMENTS

Reviewer #1	Response
1. My major concern is that about half of the included studies (12 studies, 1282 patients) had zero deaths and were therefore actually “excluded” (given zero weight) in the meta-analysis. There were also many studies with very few event numbers and were given limited weight in data combination. These studies often focus on mild/moderate COVID-19 patients or outpatients, whereas the RECOVERY trial (with 73.7% weight in meta-analysis) included a large number of severe hospitalized patients. Although this review intended to combine results from both inpatients and outpatients, as well patients of various disease severity, it actually shifts towards severe hospitalized COVID-19 patients when all-cause mortality was the only outcome of interest. This limitation shall be carefully addressed in the manuscript.	Thank you for this important comment, which we have now carefully addressed by adding a specific limitation to our discussion: “Third, although this analysis intended to combine results from both inpatients and outpatients regardless of disease severity, trials enrolling patients with mild to moderate disease comprised a minority of the final sample size; many of which had zero or few events.” We now also highlight this in the Results section and state “In 12 trials including a total of 1282 patients (representing 12.8% of the total sample size for HCQ), there were zero deaths in both arms”.
2. Although mortality is the only outcome focused in the review, I would suggest also briefly discussing potential effects of HCQ in other outcomes, e.g. possible reduction in viral load, especially when many included studies did not have mortality data.	While we agree that other outcomes than mortality may be of relevance, we did not collect these data as this information (such as viral load) would be difficult to be obtained based on aggregated and non-harmonized trial information in a rapid approach as ours, in particular for ongoing trials. It would also

Alternatively, a narrative summary for additional outcomes may be performed for those studies with zero deaths.

require a careful review and specific assessment of further biases which would be far beyond the scope of this rapid meta-analysis. We wanted to focus on the most central outcome guiding decision-making, which is also the least affected by biases. We believe a non-systematic discussion of this minor fraction of trials with zero events (12.8% of the total sample size for HCQ) would not be compatible with our prespecified systematic process. Please also note (as shown in Table 1) that several trials without mortality data were very small (7 of them have less than 35 patients) and we believe a brief summary would not be adequate and not allow reliable statements about effects on e.g. viral load.

However, we have further highlighted this limitation and now state:

“We did not review individual trials, nor did we stratify results according to patient characteristics, and we have not collected information on other outcomes than mortality. Such analyses are planned in future publications using in-depth details disclosed in individual trial publications to come.”

3. Studies allocating patients with confirmed or suspected SARS-CoV-2 infection were included. It would be nice if studies with suspected SARS-CoV-2 were marked in Table1.

Thank you for this suggestion; we have added the information in Table 1 accordingly.

4. There was no assessment for publication bias of the included studies. Funnel plots or Egger’s test should be performed.

We have now added a funnel plot (Figure S1) to complement our discussion on potential publication bias as suggested.

We now say in the Methods: *“We also assessed small-study effects with an inverted funnel plot and Egger’s test. (Higgins, J. P. T. et al. Cochrane Handbook for Systematic Reviews of Interventions. (John Wiley & Sons, 2019)). Presence of small-study effects may be suggestive, but not definitive, of publication bias.”* and state in the Results: *“We identified no relevant risk of bias across all trials, with only one trial including 7 patients having an overall high risk of bias (Table S2). We found no evidence of small-study effects (Figure S1).”*

We would like to stress that in this initiative, we made a thorough effort to include unpublished work.

The funnel plot emphasizes this point by illustrating published and unpublished trials.

Reviewer #2

The statistical methods seem robust, and the conclusions may well be valid, the authors discuss the issue of reporting bias in relation to the included trials, but do not discuss the potential impact of the unavailable trials on their results. While those that are still recruiting may struggle to do so now, those that have completed or been discontinued should have results that could contribute. It strikes me that they may not have heard from some investigators because their trials had negative or null results, in which case this might be an optimistic view of the effects of HCQ/CQ.

We agree that this is an important point and now specifically investigate potential publication/reporting bias by adding a funnel plot (please also see response to Reviewer 1). We have found no indication for relevant publication or reporting bias.

Moreover, the remaining studies are likely too small to have a substantial impact on our findings (please see table of excluded studies) and it is also highly unlikely for trials to have actually reached their target sample size.

We checked the registry entries and saw that only 5 are marked as “terminated” (i.e. discontinued after recruitment started). Their combined target sample size is 2485 participants. The biggest of those 5 trials, with a targeted size sample size of 1660, had only recruited 148 patients, i.e. only a fraction of the total targeted sample size of the 26 trials that we have included for HCQ. To further clarify this, we now describe the trial status in Table 2 a bit more detailed (and we report the numbers according to the date of the last search, for consistency).

Another omission is any assessment of the ‘quality’ or risk of bias of the included trials. As stated in the protocol, all cause mortality may limit detection biases and they describe blinding in the text, but what about the potential for other biases? This is important particularly for unpublished trials and data, which at the moment need to be taken at face value. I understand that time is of the essence, but even a basic assessment of e.g. , the randomisation process, would re-assure the reader as to the quality of the trials, and evidence presented. This should be straightforward for published trials and presumably, for unpublished trials, could be done on the basis of trial registration data together with investigator input. I don’t think it is valid to postpone this a future

We have followed this suggestion and now report the assessment using the Cochrane Risk of Bias (RoB) 2.0 instrument). As suggested, we used the trial registration information and specifically assessed the randomization process requesting information from all trialists.

We now report: *“We identified no relevant risk of bias across all trials, with only one trial including 7 patients having an overall high risk of bias (Table S2). We found no evidence of small-study effects (Figure S1).”*

publication.

Was all-cause mortality measured at same time point e.g. 28 days in all trials? If not, could it have a bearing on the results?

We did not restrict the mortality reports to the same time point in order to gather all available data, and not to lose trials because of different follow-up. This was not clear in the manuscript and only stated it in the protocol, thank you for highlighting this. We have now made a clarification and state:

“We included trials regardless of follow-up time and whether mortality was a primary outcome or not; moreover, we put no restrictions on trial status, language, geographical region, or healthcare setting.”

We have no reason to assume that the effect of HCQ on all-cause mortality would be different when measured at different time-points before or after hospital discharge, and therefore think that our findings are not affected by this issue.

I find the main manuscript text, tables and the figures in particular to give too much of a meta-analysis methodology perspective (e.g. publication status, comparing models, cumulative meta-analysis). Although these add to the interpretation and should be referred to, it would be better if the data presented was geared to providing clinical insight? For example, I would suggest grouping table 1 by drug, then setting, and control group to be more important than published vs unpublished. Also, it would be good to include at the very least HCQ subgroup plots by setting and control comparator as well as the published versus not in the main manuscript. For these, it is important to see the data (events/pts) on the plot (not just table S2A). Given that a few trials contribute most of the the cumulative meta-analysis plots are less informative, and so could be relegated to the appendix.

Thank you very much for your suggestion.

We changed the ordering of the table accordingly to a more clinical perspective. Table 1 has been reordered as suggested by HCQ and CQ, patient setting, and then control type. Additionally, we added a column reporting the COVID-19 status of included patients.

We agree that a potential effect modification across different subgroups is of interest. We have now added forest plots stratifying trials for patient setting and control type in the supplement. However, we prefer to show those plots in the supplement, as none of the interaction tests supports the hypothesis of subgroup effects and we do not want to overemphasize this issue. We agree that a few studies contribute most to the results. However, for this very reason, we believe it is important to keep the cumulative meta-analysis in the manuscript. A key point of this work, with this special and novel approach, was to analyze the development of the evidence in the context of the COVID-19 pandemic, and to illustrate the influence of small and partly discontinued studies on the existing evidence.

The discussion could be better structured better to make the messages clearer, for

Thank you for helping us structure the discussion. We considered adding subheadings but noted that author

example: key findings, context, strength, limitations and implications.

guidelines of the journal advise against this use. Between “context” and “strengths and limitations” we discuss characteristics relating to internal validity, generalizability, and evidence generation. We now restructured the discussion so that the two paragraphs related to evidence generation (publication bias and recruitment success) are adjacent.

Reviewers' Comments:

Reviewer #1:

Remarks to the Author:

The authors have kindly addressed all my suggestions and there is no additional comments.

Reviewer #2:

Remarks to the Author:

The authors have made a considerable efforts to respond to both sets of reviewers comments, in particular adding greater clarity and providing a more in depth investigation of the potential for bias. Also, Table 1 is much more informative.